# Micromachined Tools Using Acoustic Wave Triggering for the Interaction with the Growth of Plant Biological Systems

**DOI:** 10.3390/mi13091525

**Published:** 2022-09-15

**Authors:** Simone Grasso, Francesca Di Marcello, Anna Sabatini, Alessandro Zompanti, Maria Vittoria Di Loreto, Costanza Cenerini, Francesco Lodato, Laura De Gara, Christian Cherubini, Giorgio Pennazza, Marco Santonico

**Affiliations:** 1Unit of Electronics for Sensor Systems, Department of Science and Technology for Humans and the Environment, Università Campus Bio-Medico di Roma, 00128 Rome, Italy; 2Unit of Computational Systems and Bioinformatics, Department of Engineering, Università Campus Bio-Medico di Roma, 00128 Rome, Italy; 3Unit of Electronics for Sensor Systems, Department of Engineering, Università Campus Bio-Medico di Roma, 00128 Rome, Italy; 4Unit of Food Science and Nutrition, Department of Science and Technology for Humans and the Environment, Università Campus Bio-Medico di Roma, 00128 Rome, Italy; 5Unit of Nonlinear Physics and Mathematical Models, Department of Science and Technology for Humans and the Environment, Università Campus Bio-Medico di Roma, 00128 Rome, Italy

**Keywords:** plant biological system, acoustic wave, mechanical growth stimulation

## Abstract

A plant biological system is exposed to external influences. In general, each plant has its characteristics and needs with specific interaction mechanisms adapted to its survival. Interactions between systems can be examined and modeled as energy exchanges of mechanical, chemical or electrical variables. Thus, each specific interaction can be examined by triggering the system via a specific stimulus. The objective of this work was to study a specific stimulus (mechanical stimulation) as a driver of plants and their interaction with the environment. In particular, the experimental design concerns the setting up and testing of an automatic source of mechanical stimuli at different wavelengths, generated by an electromechanical transducer, to induce a micro-interaction in plants (or in parts of them) that produces a specific behavior (hypothesis) of plants. Four different experimental setups were developed for this work, each pursuing the same objective: the analysis of the germination process induced by stimulation by sound waves in the audible range. It can be said that the introduction of sound waves as a stimulant or a brake for the growth of plants can offer significant advantages when used on a large scale in the primary sector, since these effects can be used instead of polluting chemical solutions.

## 1. Introduction

System biology is a scientific discipline whose research activity, intrinsically multidisciplinary, is devoted to detecting, studying, and interpreting interactions within biological systems with a holistic approach [1]. Focusing on plants, a “plant system” is exposed to external agents. In general, each plant has its features and needs with specific interaction mechanisms suitable for its survival. A list of parameters can influence biological systems: chemical, biological, and physical quantities [2]. Interactions between systems can be studied and modeled as energy exchanges of mechanical, chemical, or electrical quantities [3]. Thus, any specific interaction can be studied by triggering the system via a specific stimulus.

The aim of this work was to investigate a particular stimulus (mechanical stimulation) as a driver for plants and their interaction with the environment. In particular, the experimental design involved the setup and testing of an automatic source of mechanical stimuli at different wavelengths generated by an electromechanical transducer to induce a micro-interaction in the plants (or in parts of them) that induces (hypothesis) a specific behavior of the plants.

Sound vibrations, intended as mechanical sources, represent a significant stimulus not only for the growth and development of the plant but also for positively influencing the germination process. This hypothesis has been corroborated by many studies that have explored this topic in depth over the last two decades [4,5,6,7,8].

Ultrasound was successfully studied and evaluated as a priming technique in different seed species, effectively producing an increase in the germination rate. However, long periods of exposure to these frequencies can have a negative effect in terms of increased germination. They can also induce mutagenesis, even if these manifestations depend on the type of ultrasound device used and the type of seed treated [9].

On the contrary, using sound vibrations at audible frequencies proved a great advantage in increasing germination. Frequencies higher than 70 Hz increased the germination speed of the Arabidopsis thaliana species [10].

Furthermore, the biological effect of sound waves at a frequency of 0.4 kHz and an SPL of 106 dB on paddy seeds was evaluated by observing a significant increase not only in the germination index but also in the height of the stem, as well as in the new weight rate (*p* < 0.01) and root system activity, and improved cell membrane permeability (*p* < 0.05) [11]. However, significant inhibition of the processes mentioned above was found in rice seeds when the sound wave stimulation exceeded 4 kHz or 111 dB [11].

On these bases, the work presented here had the aim of investigating the effects of multi-frequency acoustics waves on the germination process. Specifically, the goal was to demonstrate that sound waves in the frequency range between 300–500 Hz and 3–5 kHz can favor the germination of seeds such as chickpea, green bean (type dwarf green), borlotto bean, wheat, lentil, pea and rice. Seed selection was made by considering foods whose consumption is widespread and which, from a nutritional point of view, represent an excellent source of both macronutrients and micronutrients, such as iron, potassium, magnesium, zinc, fiber, folate and antioxidants [12]. Furthermore, their use implies the correct exploitation of soil in terms of biodiversity [13]. Their worldwide production and consumption have also increased in recent years and are expected to increase until 2029 [14]. In this context, to cope with a growth in demand and the greater awareness of consumers in terms of health and food and environmental safety, even scientific research, in this sense, must aim at solutions that are as sustainable as possible to increase and encourage world production.

Could it be that acoustic wave interactions with plants are likely to model this approach? From a theoretical point of view, the problem of the vibrational modes of an elastic sphere free from externally imposed stresses was analytically studied by H. Lamb in 1882 [15]. This study requires an understanding of the natural free vibration modes (normal modes) of the system and its resonant states. Most of the associated literature relies on numerical simulation, although in some cases, analytical formulas, mostly in approximated forms, were found, and we adopted these for our experimental campaign. In Ref. [16] in particular, one can find an analytical high-frequency study. In detail, the approximate expression of frequencies in Hz for longitudinal and transversal waves in a free homogeneous elastic sphere of radius *R* is given by $f_{L}\approx \frac{v_{L}}{2R}\left(\frac{l}{2}+n_{L}+1\right)$, $f_{T}\approx \frac{v_{T}}{2R}\left(\frac{l}{2}+n_{T}\right)$ where $n_{L}$, $n_{T}$, and 𝑙 are integer numbers associated with the various modes. In these formulas, $v_{L}=\sqrt{\left(E/\rho \right)}$ and $v_{T}=\sqrt{\left(G/\rho \right)}$ are the longitudinal and transverse sound speeds in the medium, respectively, with *E* being Young’s modulus, *G* being the shear modulus and ρ  being density. Resonant frequencies, in this solid case, can be mostly found using numerical simulations. In Ref. [17], however, one can find an analytical study of the frequency resonances of a pulsing sphere of radius *R* filled with a liquid and harmonically forced on its boundary, resulting in $f_{n}=\left(n\;\pi \;v_{s}\right)/R$, with $v_{s}$ being sound speed and *n* = 1,2,3, …. It is important to stress that these frequencies have a similar mathematical structure to the previously discussed ones, differing by multiplicative numerical factors only. Here, we study free oscillations only, however. We are interested in particular in having simple theoretical hints that are useful to properly restrict the frequency domain in which we perform our experiments, so we assume: an almost ellipsoidal seed (whose semiaxes are given by *a, b*, and *c*) approximated as an elastic sphere with mean radius given by *R* = (𝑎 + 𝑏 + 𝑐)/3; pure radial longitudinal oscillations so that we have 𝑣 = $\sqrt{\left(E/\rho \right)}$ and for the sake of simplicity, 𝑙 = 1 and $n_{L}$ = 0. For a typical lentil (*a* = 3.15 mm, *b* = 3.05 mm, and *c* = 2.1 mm), we have *R* = 2.8 mm and 𝜌 = 1380 kg/m^3^ [18]. Using cereal grains’ mechanical measurements [19], we adopted the mechanical parameter *E* = 10.4 MPa. Using these values, we obtain *f*~ Hz = 23 kHz, so, due to the strong theoretical simplifications assumed, we performed experiments in the range of several kHz.

## 2. Materials and Methods

### 2.1. Transducers

Four different experimental setups were developed for this work, each with the same objective: the analysis of the germination process induced by stimulation with sound waves in the audible range. In the first experimental setup, a portable Bluetooth acoustic emitter was used, the essential characteristics of which are shown in Table 1.

In the second experimental setup, the acoustic waves were generated using two magneto dynamic transducers (TEAX19C01-8, Tectonic Audio Labs Inc., Woodinville, Washington, USA) characterized by an emitter size of 19 mm, with a nominal power of 2 W and an impedance of 8 Ω. This transducer emits sound vibrations between 300–500 Hz and 3–5 kHz together with dedicated software for generating audio tracks. The transducers are connected via an amplifier board and a PC equipped with a 1.4 GHz 64-bit quad-core processor, 2.4 GHz wireless LAN and 5 GHz IEEE 802.11.b/g/n/ac, one full-size HDMI, 4 USB ports 2.0, a 5 V/2.5 A DC power input and a CSI camera port. In this case, the camera (IMX219, SONY, Tokyo, Japan) is connected to the single-board computer, and it is characterized by a resolution of 8 megapixels with a maximum capture rate of 30 fps. Furthermore, it allows high-definition (HD) pictures to be taken by providing less image contamination, such as fixed noise of the patterns and smudges. This device was remotely monitored through dedicated software. The audio tracks were generated by Audacity software (v3.13, MuseCY Holdings Ltd., Limassol, Cyprus), an editor and recorder for multitrack audio. By using CronTab, it was possible to plan the automatic execution of a script relating to the duration of the sound reproduction and the acquisition of an image. The CronTab command allows the establishment of scheduled operations, i.e., commands executed periodically in a completely automated way by the operating system.

### 2.2. Sample Plants

The experiments tested seven types of seeds: (1) chickpea, a wrinkled beige-yellowish seed variety; (2) green bean (type dwarf green “mangiatutto”), a very productive variety with high resistance to high temperatures; (3) borlotto bean, a climbing variety that grows well even at low temperatures; (4) wheat; (5) red lentil; (6) pea, a late-cycle variety that grows well in almost all soils; and (7) brown rice.

Many seeds possess an endosperm, which plays a fundamental role in supporting the embryo’s growth, providing the nutrients it needs. Recent advances in the biology and physiology of seeds have shown that the endosperm can respond to environmental signals, thus regulating the embryo’s growth. Thus, the germination process represents a systemic response involving bidirectional interactions between the germ and endosperm. Wheat and rice possess this structure. On the contrary, exalbuminous seeds, mainly represented by legumes such as chickpeas, green beans, kidney beans, lentils, and peas, do not possess this structure and reserve function.

### 2.3. Experimental Setup

Four different experimental setups were used to perform four different experiments.

#### 2.3.1. First Experiment

A total number of 40 rice seeds, 40 wheat seeds, 20 pea seeds, 10 seeds, 30 lentil seeds, 15 borlotto bean seeds and 15 green bean seeds were prepared. Each batch of seeds was placed on top of a cotton pad soaked with 40 mL of water inside a Petri dish (7 cm in diameter), as shown in Figure 1a. A duplicate Petri dish was prepared for each seed variety. Thus, the first seven represent samples treated with sound waves, and the other half are intended to be the control samples (not subject to the treatment with acoustic waves).

All of the plates, both processed and control samples, were placed inside plastic bags with hermetic closure to recreate an environment with relative humidity values of about 90%.

For mechanical wave stimulation, the intensity of the transducer was set to its maximum. It was connected via a USB cable to an Asus computer (model PU301L, ASUSTeK Computer Inc., Taipei, Taiwan), equipped with its integrated sound card, which was set at an intensity level of 80%. Air was the only medium: the transducer was placed at 15 cm from each sample, as outlined in Figure 1b. The system was composed of seven plates, on which the seeds were exposed to the mechanical wave stimulus treatment using a Bluetooth transducer, connected through a USB cable to the computer described above to select the audio track to be played.

In this case, original Javanese music was used, particularly the “Puspawarna” track. Recent studies have applied this kind of music to plants at characteristic frequencies of 3–5, 7–9 and 11–13 kHz for a time-variable exposure of 1, 2 and 3 h. Positive results were found for plant growth, with increased productivity for the frequency range of 3–5 kHz and an exposure time of 2 h [20]. However, for this experiment, no specific frequencies were tested, but the seeds were stimulated with waves of multi-frequency sound for three consecutive hours once a day (at 10:00 am).

Twice a day for a week, in the morning at 10:00 and late afternoon at 18:00, the data relating to the total germinated seeds were collected for both control and seed plates treated with acoustic vibrations, and pictures of the samples were acquired.

#### 2.3.2. Second Experiment

The second experiment was carried out on a duplicate of red lentil samples. First, the seed growth media were prepared: cotton was placed to cover their entire surface on two Petri dishes, with a square shape of about 120 mm per side. Both cotton and Petri dishes were sterilized with ultraviolet radiation for two hours together with other tools used for sample preparation, including a laboratory falcon, a graduated conical-bottom tube with a screw cap, and 50 mL and 250 mL laboratory bottles containing distilled water.

All of the operations were carried out under the hood using sterilized latex gloves. Three lab spoons of red lentils were collected and transferred to the graduated conical test tube with 30 mL of ethanol. The tube was stirred for two minutes, at the end of which the ethanol was discarded. Then, distilled water was added to the test tube, and a series of washes (about seven) was performed to eliminate ethanol residues to avoid seed damage. The seeds were then arranged in 7 rows of 7 seeds each for 49 red lentil seeds per plate (Figure 2).

The same operations described above were repeated both for the samples treated with mechanical waves and for the control samples. The two plates with the lentil samples were placed inside a growth chamber with controlled temperature and brightness. The temperature reached values of 22–23 °C. Brightness depended on the position of the two plates (Figure 2): specifically, the left plate was irradiated with 6100 lux, while the right one was radiated with 5750 lux. For this reason, the plates were placed on an orderly grid with letters and numbers to precisely define any location. The photoperiod was set to cycles of 16 h of light and 8 h of darkness. The plates rested on a raised plastic support made with a 3D printer (Ultimaker S3, Ultimaker B.V., Utrecht, Netherlands). This left valuable space to allow the fixation of the magneto dynamic transducers in a central position on the lower portion of the plates, as shown in Figure 3a.

Figure 3b shows the locations of the various devices and samples. Acoustic vibrations stimulated the samples with frequency values of 300–500 Hz or 3–5 kHz every hour for 450 s (7.5 min). In addition, the transducer’s intensity was set to a power of 1.8 W. The entire experiment had a total duration of twenty-one days: first week, 300–500 Hz; second week, 3–5 kHz; and third week, the absence of stimulation. During the experiment, on the first five days of the week, twice a day (in the morning at 9:30 and the afternoon at 17:30), the cotton of all plates (controls and stimulated ones) was soaked with 20 mL of water. At the end of each week, the following data were collected: the total number of germinated seeds; the total number of seedlings grown; and the weight and height of each plant.

#### 2.3.3. Third Experiment

The experimental setup just described (second experiment) was also followed for the third experimental setup. Therefore, the same device was used to generate and reproduce sound vibrations. Similarly, for the growth media, all of the steps for their preparation were the same. However, 40 wheat seeds were tested but without any pretreatment. This experiment was conducted on duplicates of Petri dishes in the same conditions in the growth chamber, and it had a duration of 21 days. During the first week, the samples were treated with sound frequencies in the range of 300–500 Hz; in the second week, frequencies were set at 3–5 kHz; and finally, during the third week, the germination process was evaluated for the control sample. During the experiment, on the first five days of the week, twice a day (in the morning at 9:30 and the afternoon at 17:30), the cotton of all plates (controls and stimulated ones) was soaked with 20 mL of water. At the end of each week, the following data were collected: the total number of germinated seeds; the total number of seedlings grown; and the weight and height of each plant. Those values were recorded in a spreadsheet to quantify and graph the trends of the germination and growth of shoots.

#### 2.3.4. Fourth Experiment

In the fourth experimental setup, the devices for the generation and reproduction of acoustic waves were the same as those used both in the second and in the third experiments. The seeds under test were: chickpea; green bean (type dwarf green mangiatutto); borlotto bean; and pea.

Seven seeds were selected for each variety listed above, and the dynamics of hydropriming was studied, meaning the time necessary to reach the maximum soaking value. The weights of the dried seeds were first measured using an analytical laboratory balance. Afterwards, they were placed to soak in a glass Petri dish of about 15 cm in diameter and placed on a sheet of paper on which a grid was drawn that separated both the single seeds and the different varieties of seeds, as shown in Figure 4.

Periodically, every thirty minutes, all of the seeds were taken from the water, dried thoroughly with absorbent paper and weighed again using an analytical laboratory balance, taking care to note the weights of the single seeds for each measure. The operation was repeated until all seeds had absorbed the maximum amount of water. Once the optimal soaking time had been established, two samples that were the same in terms of variety and number of seeds were prepared. An example includes: 16 pinto bean seeds, 16 chickpea seeds, 16 green bean seeds (type dwarf green mangiatutto) and 16 pea seeds. Preliminarily, the initial weights of the two samples’ seeds were measured with a laboratory analytical balance. Subsequently, as schematized in Figure 5a,b, both samples were immersed in water inside two different Petri dishes made of glass and placed in two different environments. One sample represented the control, while the other was the stimulated sample. Sound waves were transmitted by direct contact between the magneto dynamic transducer and the Petri dish containing the water and seeds. The sound treatment was continued for ninety minutes. At the same time, the seeds of the control sample were also immersed in water inside the second Petri dish; however, in this case, there was no stimulation.

At the end of the treatment, after ninety minutes, the seeds of the control and sound wave-treated samples were dried thoroughly with absorbent paper and weighed again. Thus, for each experiment, the average weight was compared before and after soaking to find any differences between the treated and control samples and then evaluate the possible effects of sound on the hydropriming process. Finally, all seeds were planted on cotton, placed inside plates of Petri dishes and divided into processed and control samples. Both plates were stored in the growth chamber with a temperature of 22–23 ° C and controlled brightness without any further acoustic stimulation.

## 3. Results

The results obtained from the four experiments are reported in the following four subsections.

### 3.1. First Experiment

The first experiment evaluated the effects of high-frequency sound waves on germination. Different seed species were compared, considering treated and control samples. To better understand the behavior, the percentage of variation in germination was evaluated and compared among the different conditions. The germination trends over time of the treated samples of rice seeds, wheat, pea, chickpea, lentil, green bean and borlotto bean are reported in Figure 6.

Lentil and wheat seeds reached almost 100% germination after 80 h, followed by the green bean, whose maximum germination values after 80 h reached about 87%. Lower percentages were obtained for borlotto bean, chickpea, and pea chutney germination: 40%, 30% and 27.5%, respectively. Conversely, rice did not show any particular trend in the process of germination. The line graph drawn in Figure 7 shows the trend of the germination percentage of the control samples.

Again, lentil and wheat seeds reached almost 100% germination after 80 h. The same results for the percentage of germination also occurred for peas, which had the same value equal to 86.7%. The seeds of chickpea were below 50% variation, as were borlotto beans, green beans, and rice. Like the treated sample, the latter did not show a trend in the germination process.

These first results show no significant differences between the treated and control samples, probably because the seeds were stimulated exclusively with multi-frequency sound waves and not in a specific range of frequencies. Moreover, the experiments were conducted in a domestic environment with uncontrolled brightness and temperatures. Such parameters are fundamental in the germination process, especially in its initial stages, affecting it positively or negatively.

### 3.2. Second Experiment Results

In the second experiment, mechanical waves with frequencies in the range of 300–500 Hz and 3–5 kHz were applied to the germination process of lentil seeds. As in the first experiment, the results were compared with the control samples.

To standardize the methods for this experiment, 49 seeds were considered. Considering the average of the two plates, it can be noted that the fresh weight, or the weight of the shoot expressed in milligrams (mg), was more significant in the treated ones. In fact, in the seeds treated with 300–500 Hz, this value reached 11 mg; in seeds stimulated with frequencies between 3–5 kHz, approximately 9 mg of fresh weight was reached, while the sample values of the control stood at only 4 mg. These results are reported in Figure 8, where the control’s weights are represented in blue, the 300–500 Hz treated sample’s weights are shown in orange, and the 3–5 kHz treated sample’s weights are in gray.

The significance of the different weights is confirmed by the *p*-value between the control and samples treated at 300–500 Hz, which is equal to 1.5171 × 10^–9^, while between the control and samples treated at 3–5 kHz, it is equal to 1.8777 × 10^–5^. The values are far below 0.01; therefore, the difference between treated and control samples in both cases is statistically significant. In addition, the growth in the height of the shoots showed significantly higher values in the treated samples than in the control sample. In particular, frequencies in the 300–500 Hz range had a more appreciable effect than the control. These results are reported in Figure 9.

In addition, in this case, the *p*-values were calculated: 2.5 × 10^–4^ for 300–500 Hz, and 0.025 for 3–5 kHz.

Finally, we report the trend of germination percentage over time for both the control samples and the samples treated with frequencies of 300–500 Hz and 3–5 kHz, as shown in the graphs in Figure 10.

In Figure 10, it can be observed that 170 h after sowing, the samples treated with frequencies equal to 300–500 Hz reached germination percentage values equal to 81.63%, while samples processed with frequencies of 3–5 kHz reached 85.71% germination. In both cases, the percentages are different from those of control samples, which account for values of less than 80% or equal to 74.49%.

### 3.3. Third Experiment Results

In the third experiment, mechanical waves with frequencies in the range of 300–500 Hz and 3–5 kHz were applied to the germination process of wheat seeds. The results were compared with the control samples.

In this case, 40 seeds for each plate were considered. The sample treated with frequencies of 3–5 kHz had a higher total number of germinated seeds and a higher percentage of germination values than the control sample. The same results were obtained for the total number of leaflets (shoots). The total number of leaflets in the samples treated with frequencies of 3–5 kHz exceeded the total number of leaflets of the control. Considering the average of the two plates, it can be noticed that there is a different phenomenon concerning the results obtained in the second experiment for fresh weight, i.e., the weight of the sprout expressed in milligrams (mg). Indeed, in the seeds treated with frequencies of 300–500 Hz, the value of fresh weight is 48.7 mg; in seeds stimulated with frequencies of 3–5 kHz, we obtained a value of about 39.5 mg of fresh weight, while for the control samples, this value is 51.8 mg. This means that the control had, on average, a higher fresh weight than the two treated samples. What has just been described is graphically evident in Figure 11, where the control is represented in blue, the samples treated with 300–500 Hz are in orange, and then the samples treated with 3–5 kHz are in gray.

Comparing the samples processed at frequencies of 3–5 kHz with the control and those stimulated at 300–500 Hz, we obtained a *p*-value considerably lower than 0.05 (respectively, 5.23 × 10^–5^ and 2.19 × 10^–3^). Conversely, the difference between the control and treatment at 300–500 Hz is not statistically significant, with a *p*-value of 0.27. In this case, probably, the endosperm cap provides mechanical strength that can affect the entire experiment.

The growth in the height of the shoots was also significantly lower in the treated samples than in the sample control for this case (Figure 12). In addition, regarding the height of the shoots of the three samples, the *p*-value values show statistically significant differences only between the acoustic treatment with 3–5 kHz waves and the other two samples, i.e., the control and stimulation at 300–500 Hz. The presence of the endosperm in the wheat seeds can represent a brake for the phenomena observed.

Finally, we report the trend of the germination percentage over time both for the control samples and for the samples treated with frequencies of 300–500 Hz and 3–5 kHz, as shown in the graphs in Figure 13.

In Figure 14, we can observe that 170 h after sowing, the average of the two plates behaves in different ways depending on the sample examined. Samples treated with frequencies equal to 300–500 Hz reached similar percentage values of germination at 92.50%, whereas samples processed with frequencies of 3–5 kHz reached 98.75% germination. For the control samples, however, values were found to be equal to 93.75%. This suggests that, as far as germination is concerned, the optimal frequency with which to treat the grain is 3–5 kHz.

### 3.4. The Results of the Fourth Experiment

In the fourth experiment, the optimal time of hydropriming, or the maximum amount of water a seed can absorb, was evaluated. This process was carried out on chickpea, green bean, borlotto bean and pea seeds. Figure 15 and Figure 16 show the weight trend, expressed as a percentage as a function of time.

The scatter plot shows that the optimal soaking time of seeds is 90 min. This time is the result of two observations.

In the first observation, what happens before 90 min was studied. In this case, the weight increased significantly with time. The first derivative obtained from the comparison between two successive time points assumes values greater than 1 in most cases.

In the second observation, the behaviors from 90 min onwards were studied. The first derivative between two successive time points is less than 1. Thus, the weight did not grow proportional to time. In other words, after 90 min, water absorption by the seed became minimal.

Figure 15a and Figure 16a show that the weights of the seeds of the samples that were selected were very similar, regardless of whether they belonged to the control group or the treated one.

From the graphs (Figure 15b and Figure 16b), no significant differences are evident between the processed samples and the control samples. Subsequently, all of the seeds were planted, as evidenced by the description of the fourth experiment.

Figure 17 and Figure 18 report the percentages of germinated seeds and the difference between the treated sample and the control ones. Interestingly, while the effect of the acoustic treatment was negligible during the soaking of the seeds, the subsequent germination seemed to be influenced by it. In particular, as shown in Figure 18, the stimulus at a frequency of 3–5 kHz produced a higher germination rate compared to the control, while the stimulus at a frequency of 300–500 Hz gave rise to the opposite effect, as shown in Figure 16.

## 4. Discussion and Conclusions

In this work, we outlined the first evidence on the effects of acoustic waves on germination. The literature provided a general overview, starting from the characteristics and physical behavior of acoustic sound waves and illustrating the relationship between sound and plant organisms. These provide a solid basis for a more complete understanding of the topic addressed. From this point of view, a specific apparatus based on mechanical wave generation was designed in order to define the relationship between sound and plant organisms, as these have been able to provide a solid basis for a more complete understanding of the topic addressed. This apparatus was applied in four different experiments. The methods are based on careful and in-depth research that highlights the effects of acoustic vibrations in the stimulation of seeds’ benefits and disadvantages on germination. A further result achieved was that of verifying, also through numerous bibliographic contributions, that the germination trend was somehow influenced by sound stimulations with frequencies included in the ranges of 300–500 Hz and 3–5 kHz. In this particular context, it was noted that different types of seeds, including chickpea seeds, green beans, borlotto beans, wheat, lentils, peas and rice, responded differently to stresses provided by a magneto dynamic transducer. Taken as a whole, this work provides novel results. It is shown that frequencies of 300–500 Hz considerably encourage the percentage of germination of lentil seeds. A statistically significant difference was noted in terms of both fresh weight and shoot height for treated specimens compared with the control sample. This difference is most evident for the treatment with 300–500 Hz frequencies compared to 3–5 kHz. The exact frequency range had different results for the wheat seeds: in this case, it was noted that the sound stimulus had adverse effects regarding fresh weight and shoot height.

In particular, there was a statistically significant difference only for frequencies of 3–5 kHz compared to the samples treated with 300–500 Hz and the control. The data concerning the percentage of germination assume that the optimal frequency with which to treat the grain is 3–5 kHz since the latter revealed higher yields than the one treated with 300–500 Hz and the control. In addition, the action of acoustic waves at 300–500 Hz and 3–5 kHz as possible mechanical perturbations of the integuments’ external parts of the seed was evaluated. From the results reported in the fourth experiment, this hypothesis was not proven. Indeed, it is highlighted that the average weight before and after the soaking of treated samples and control samples did not vary significantly. Interestingly, the 3–5 kHz frequencies seem to positively affect the germination rate of seeds exposed to the acoustic waves during hydropriming, while the 300–500 Hz frequencies produce the opposite effect.

Furthermore, considering the experimental arrangement described in the methods section, the effect of the acoustic waves on the seeds was shown to be independent of the distance they were from the center of the Petri dishes. Indeed, the influence of the acoustic stimulus on the seed growth was found to be comparable among all of the seeds on the plate, independently of their position: this behavior is also theoretically confirmed by some other papers [21,22].

Considering the “pressure” acting on the seeds, according to theoretical and experimental studies in the field [21], in order to determine an estimate of “Loudness”, described as the sound pressure level (SPL) and measured in dB (decibels) at a certain distance, the Inverse Square Law can be used. In this case, the SPL decreases inversely with the square of the distance from the source at a rate of approximately 6 dB for each doubling of the distance: 𝑆𝑃𝐿(𝑑) = 𝑆𝑃𝐿(0) − 20𝑙𝑜𝑔10(𝑑), where 𝑑 is the distance in meters, and 𝑆𝑃𝐿(0) is the sound pressure level at one meter from the source. Taking 𝑆𝑃𝐿(0) as zero, we can obtain the expression of change in 𝑆𝑃𝐿 for distance 𝑑 as ∆𝑆𝑃𝐿(𝑑) = −20𝑙𝑜𝑔10(𝑑). Using this formula for d = 1 cm (0.01 m):𝑆𝑃𝐿(𝑑) = −20 Log10(0.01) = 40 dB

According to this, the influence of SPL on seeds in the experimental arrangement of this work is negligible. This shows that the frequencies applied had no mechanical effect on the integrity and physical structure of the seed, at least in the initial phase. Rather, it can be assumed that sound waves at frequencies of 300–500 Hz and 3–5 kHz act by influencing the biochemical and molecular behaviors of the seeds or that the sound treatment did not affect the behavior of the seeds in any way, probably due to a too short duration of stimulation. In summary, this paper tries to give continuity to efforts that emerged from the literature to address the needs that characterize the agricultural sector by reporting non-trivial evidence using a dedicated apparatus to generate and monitor the phenomena. Therefore, it can be said that introducing acoustic waves as a stimulus or brake for plant growth can provide essential benefits if adopted on a large scale in the primary sector, as these effects can be used instead of polluting chemical solutions.

In this work, the authors had the aim of identifying a novel approach in which MEMS can be applied. In particular, the new perspective on applications of MEMS includes precision biological farming. In this work, the authors propose a methodology to identify the specifications of a new transducer and acoustic sensors for MEMS. The aim of this approach was to understand whether specific frequencies can be applied in the context of biological growth systems in order to develop a suitable system. This study can offer important suggestions to researchers who work in this field. A multidisciplinary study has been arranged and conducted, putting together expertise in electronics, sensors, physical modeling, biotechnology, and plant physiology. This work could represent a basis for developing new MEMS sensors in the context of precision farming.

Different sensors can be applied on seed clusters to improve the productivity of the farm. Microsystems can be produced and placed under the soil. The device to be designed, developed and transferred to the field should be small and noninvasive: a MEMS could represent the best solution.

## Figures and Tables

**Figure 1 micromachines-13-01525-f001:**
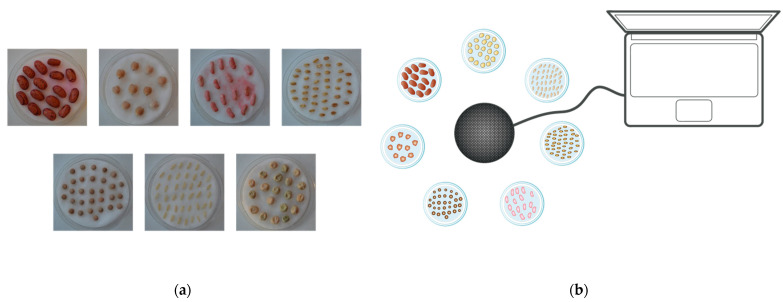
Overview of the experimental setup with the seed plates (**a**) and their arrangement relative to the wave emitter (**b**).

**Figure 2 micromachines-13-01525-f002:**
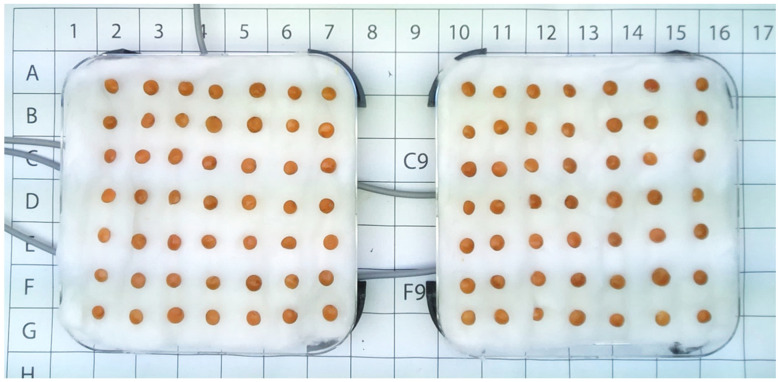
Arrangement of red lentil seeds on cotton.

**Figure 3 micromachines-13-01525-f003:**
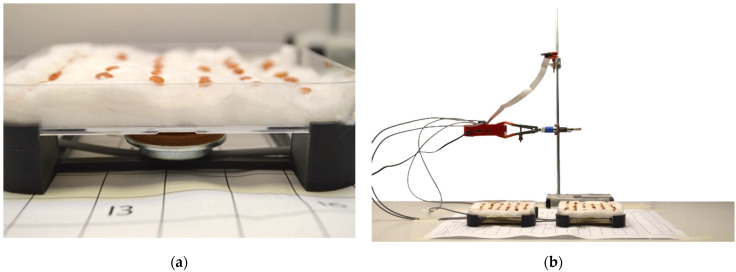
Arrangement of the plates on a plastic support in order to correctly expose the samples to the stimuli (**a**) and an overview of the experimental setup with the locations of all used devices (**b**).

**Figure 4 micromachines-13-01525-f004:**
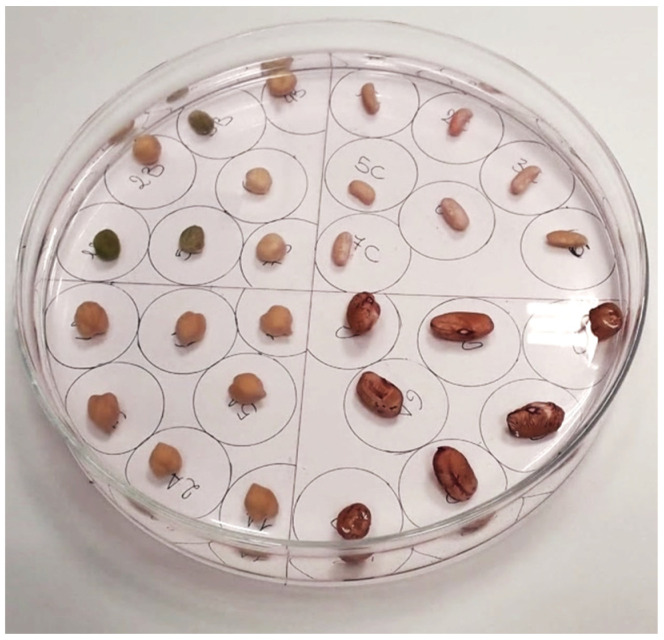
Arrangement of the different varieties of seeds on a gridded plate.

**Figure 5 micromachines-13-01525-f005:**
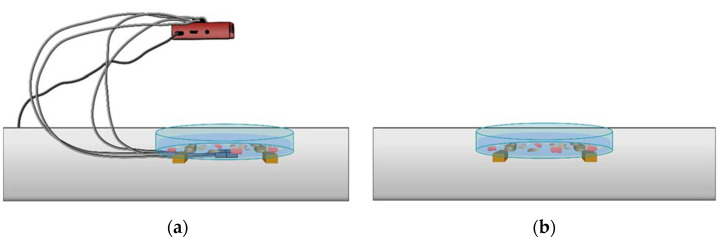
Sample under acoustic wave stimulus (**a**) and control sample (**b**).

**Figure 6 micromachines-13-01525-f006:**
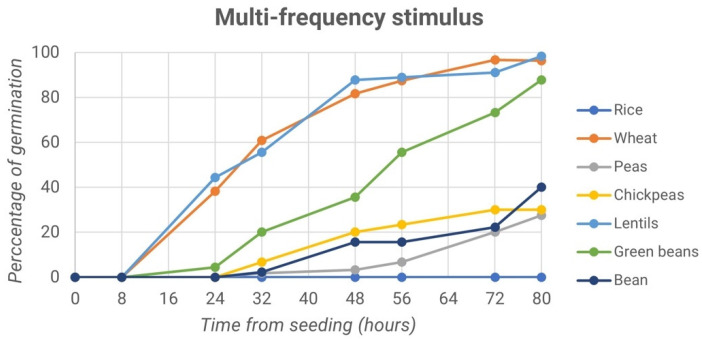
Percentage of germination vs. time from seeding: first experiment, multi-frequency stimulus.

**Figure 7 micromachines-13-01525-f007:**
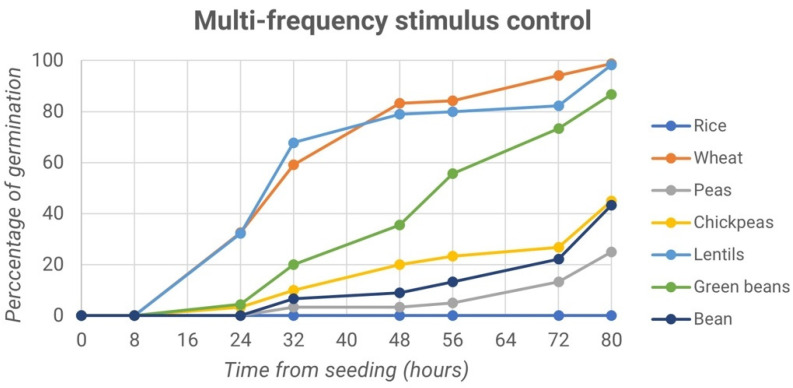
Percentage of germination vs. time from seeding: first experiment: multi-frequency stimulus control.

**Figure 8 micromachines-13-01525-f008:**
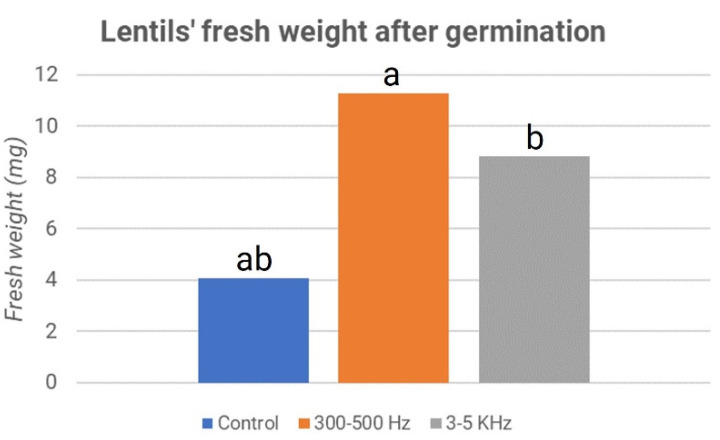
Lentils’ fresh weight after germination: second experiment. The letters “a” and “b” refer to the statistical significance of the observed difference between sample couples with a *p*-value < 0.05 and *n* = 196.

**Figure 9 micromachines-13-01525-f009:**
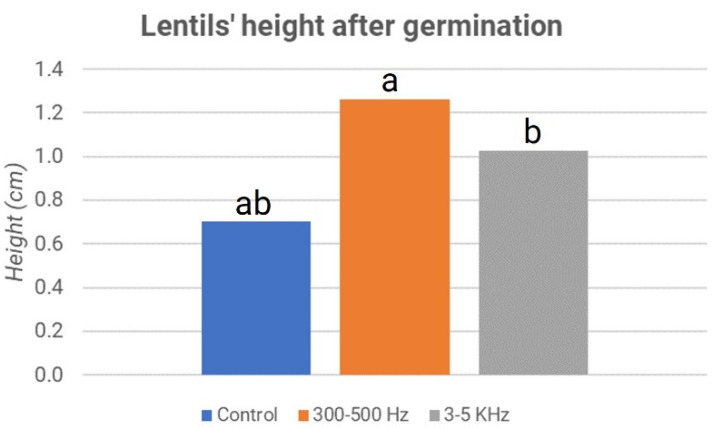
Lentils’ height after germination: second experiment. The letters “a” and “b” refer to the statistical significance of the observed difference between sample couples with a *p*-value < 0.05 and *n* = 196.

**Figure 10 micromachines-13-01525-f010:**
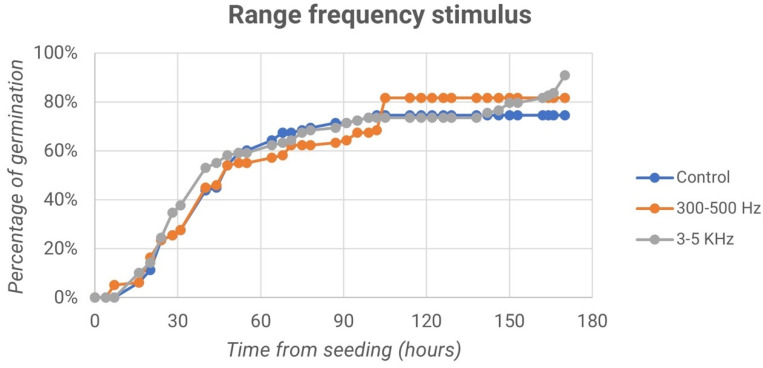
Percentage of germination vs. time from seeding for both the control samples and the samples treated with frequencies of 300–500 Hz and 3–5 kHz: second experiment.

**Figure 11 micromachines-13-01525-f011:**
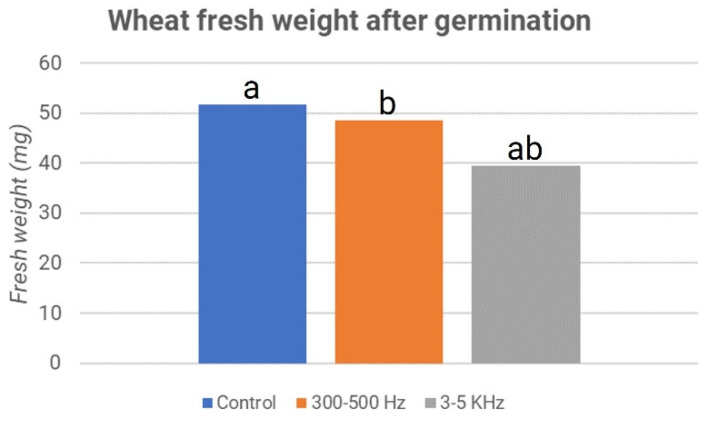
Lentils’ fresh weight after germination: third experiment. The letters “a” and “b” refer to the statistical significance of the observed difference between sample couples with a *p*-value < 0.05 and *n* = 160.

**Figure 12 micromachines-13-01525-f012:**
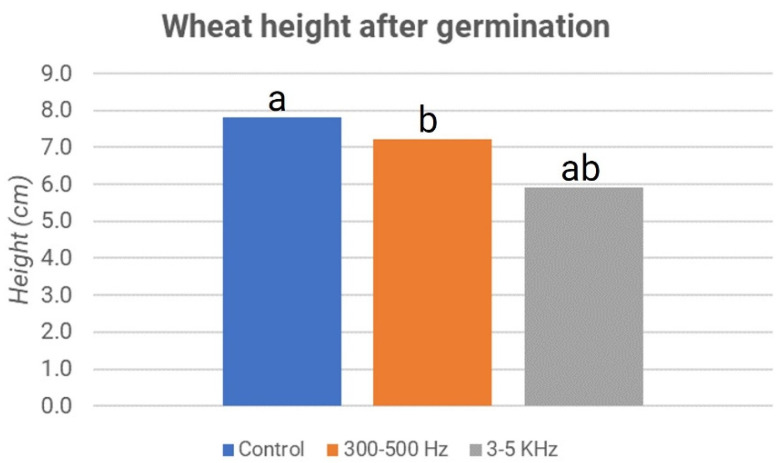
Lentils’ height after germination: third experiment. The letters “a” and “b” refer to the statistical significance of the observed difference between sample couples with a *p*-value < 0.05 and *n* = 160.

**Figure 13 micromachines-13-01525-f013:**
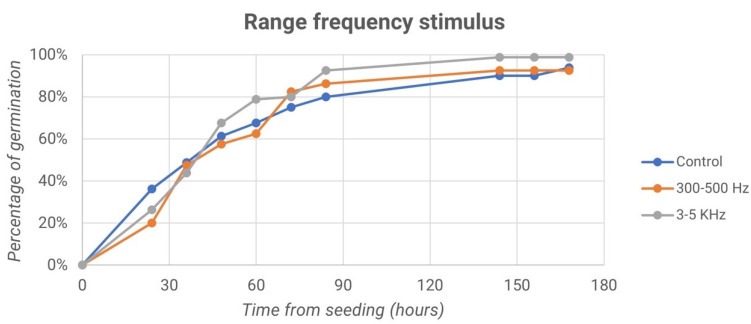
Percentage of germination vs. time from seeding for both the control samples and the samples treated with frequencies of 300–500 Hz and 3–5 kHz: third experiment.

**Figure 14 micromachines-13-01525-f014:**
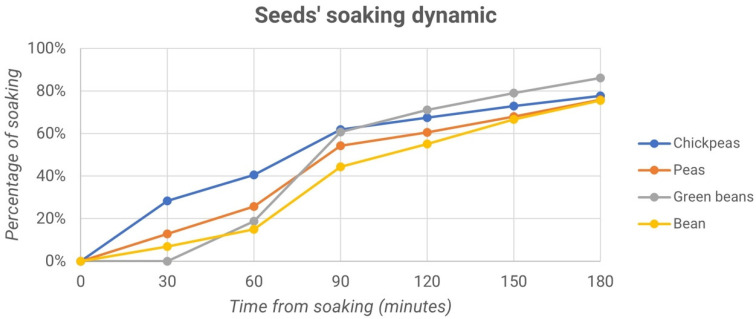
Percentage of soaking vs. time from soaking for chickpeas, peas, green beans and beans.

**Figure 15 micromachines-13-01525-f015:**
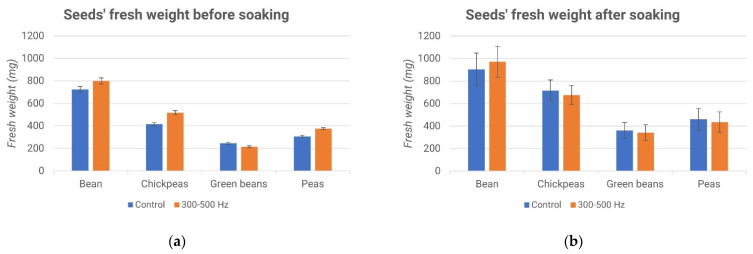
Seeds’ fresh weights before soaking (**a**) and after soaking (**b**).

**Figure 16 micromachines-13-01525-f016:**
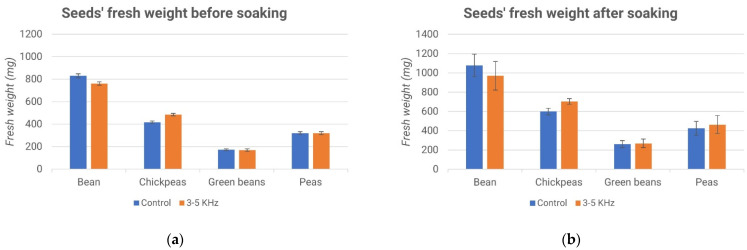
Seeds’ fresh weights before soaking (**a**) and after soaking (**b**).

**Figure 17 micromachines-13-01525-f017:**
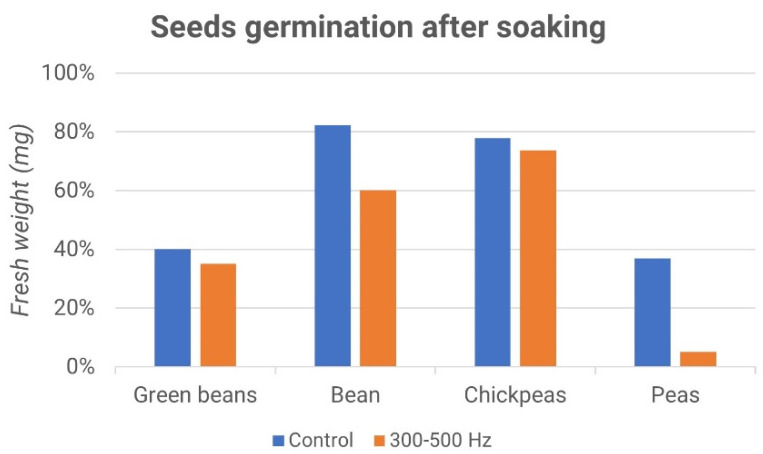
Seed germination after soaking: fresh weight vs. samples (green beans, bean, chickpeas and peas); control (blue) and 300–500 Hz stimulation (orange).

**Figure 18 micromachines-13-01525-f018:**
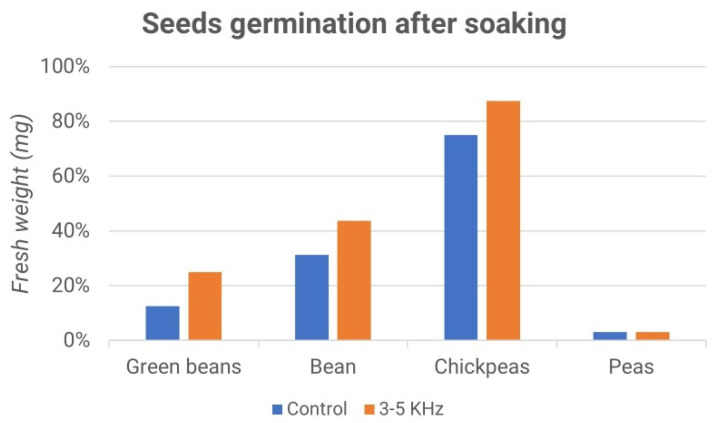
Seed germination after soaking: fresh weight vs. samples (green beans, bean, chickpeas and peas); control (blue) and 3–5 kHz stimulation (orange).

**Table 1 micromachines-13-01525-t001:** Technical characteristics of the portable Bluetooth acoustic emitter used for the generation of the acoustic waves.

Driver	
Number of transducers	1
Type of magnet	Neodymium
**Audio**	
Total harmonic distortion (THD)	1%
RMS power	3.5 W
**Technical Details**	
Height	100 mm
Width	100 mm
Depth	42 mm
Battery capacity	700 mAh
Battery type	Lithium Polymer (LiPo)

## Data Availability

The data presented in this study are available on request from the corresponding author.

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
