# Peer review of "Micromachined Tools Using Acoustic Wave Triggering for the Interaction with the Growth of Plant Biological Systems"

_micromachines, 2022, doi:10.3390/mi13091525_

Round 1
Reviewer 1 Report
In the manuscript, “Micromachined Tools using Acoustic Wave Triggering for the Interaction with the Growth of Plant Biological Systems” was submitted for publication in the journal Micromachines. The author presents a method to analyze the germination process induced by stimulation by sound waves in the audible range. The manuscript is not well prepared and written. The author needs to make major revisions and changes to some detailed information in the manuscript and resubmit it to another journal.
1. This manuscript was submitted to a special Issue "MEMS in Italy" in the journal Micromachines. However, I don’t think this manuscript matches this topic. The whole manuscript has nothing to do with MEMS. The transducer was brought from a third party. The structure and fabrication of the transducer method are unknown. Look at the scope of the special issue. This manuscript is not fit.
Here is the scope of this special issue:
“The purpose of this Special Issue is to offer an overview of the importance of MEMS in Italy, focusing on new trends in design, fabrication processes, and applications.
I therefore warmly invite you to submit contributions on all scientific and technical aspects of MEMS in Italy.
Topics include but are not limited to the following:
Theory and multiphysics working principles of MEMS;
Multiphysics modeling and simulations for MEMS;
Fabrication processes for MEMS, including additive manufacturing at the micro-scale;
Experimental characterization and reliability for MEMS;
New trends in MEMS design and production;
Innovative applications of MEMS;
MEMS low-power sensing techniques and devices.”
2. There is an issue in the research design. In Figure 2 and Figure 3, the author tried to arrange the plates on plastic support in order to correctly expose the samples to the stimuli. However, the transducer is quite small than the Petri dish. Some of the seeds are placed out of the diameter of the transducer. The acoustic wave propagated to the liquid will generate the longitude wave and volume pressure to the seeds. The acoustic force on the seeds placed at the different locations will be different. The seeds placed at the edge will expose to different acoustic forces compare to the seeds in the range of the acoustic transducer diameter.
Author Response
In the manuscript, “Micromachined Tools using Acoustic Wave Triggering for the Interaction with the Growth of Plant Biological Systems” was submitted for publication in the journal Micromachines. The author presents a method to analyze the germination process induced by stimulation by sound waves in the audible range. The manuscript is not well prepared and written. The author needs to make major revisions and changes to some detailed information in the manuscript and resubmit it to another journal.
- This manuscript was submitted to a special Issue "MEMS in Italy" in the journal Micromachines. However, I don’t think this manuscript matches this topic. The whole manuscript has nothing to do with MEMS. The transducer was brought from a third party. The structure and fabrication of the transducer method are unknown. Look at the scope of the special issue. This manuscript is not fit.
Here is the scope of this special issue:
“The purpose of this Special Issue is to offer an overview of the importance of MEMS in Italy, focusing on new trends in design, fabrication processes, and applications.
I therefore warmly invite you to submit contributions on all scientific and technical aspects of MEMS in Italy.
Topics include but are not limited to the following:
Theory and multiphysics working principles of MEMS;
Multiphysics modeling and simulations for MEMS;
Fabrication processes for MEMS, including additive manufacturing at the micro-scale;
Experimental characterization and reliability for MEMS;
New trends in MEMS design and production;
Innovative applications of MEMS;
MEMS low-power sensing techniques and devices.”
Thanks to the reviewer for the observation about the relevance of the manuscript for the special issues on MEMS in Italy. This observation allows the authors to better specify the pertinence of the work, as agreed with the Editor before paper preparation. Indeed, in this work, authors have the aim of identifying a novel approach in which the MEMS can be applied. In particular the new perspective of applications of MEMS: precision biological farming. In this work the authors, according to the editor's request, have proposed a methodology to identify the specifications of new transducer to be developed as MEMS acoustic sensors. It is true that the specifications have been obtained by using a transducer produced by a third party, but the role of this approach was to understand if specific frequencies can be applied to the contest of biological growth systems in order to develop a suitable system. This study can offer important suggestions to the researchers that work in this field. These considerations are complex because for this kind of applications multidisciplinary studies have to be arranged. The authors think that this work could represent a basis on which developing new MEMS sensors in the contest of precision farming, thus it fits the following topic reported above: “Innovative applications of MEMS” (the specific application proposed) and “New trends in MEMS design and production”, meaning the design for precision farming (which is a new trend) which requests experimental data coming (also) from this work.
Different sensors can be applied on seeds cluster to improve the productivity of the farm. The microsystems can be produced and placed under the soil. The science requires that these devices are small and not invasive and in this context, the MEMS could represent the best solution.
- There is an issue in the research design. In Figure 2 and Figure 3, the author tried to arrange the plates on plastic support in order to correctly expose the samples to the stimuli. However, the transducer is quite small than the Petri dish. Some of the seeds are placed out of the diameter of the transducer. The acoustic wave propagated to the liquid will generate the longitude wave and volume pressure to the seeds. The acoustic force on the seeds placed at the different locations will be different. The seeds placed at the edge will expose to different acoustic forces compare to the seeds in the range of the acoustic transducer diameter.
Thanks to the reviewer. The transducer is placed under the Petri plate dish to stimulate the seeds. The seeds are not immersed in a liquid but in a cotton matrix soaked with 20 mL of water. The authors understand reviewer’s doubt and arranging the experimental design have verified that the seeds at a different distance in the Petri dishes have been affected in the same way from the acoustic wave. We agree that it is mandatory to specify this detail in the text. To better explain this aspect, the seeds exposed to the stimulus are growing in the same way, and no correlation has been observed between the distance and the size of the transducer. The seeds far from acoustic source have the same probability of growth. This behavior is also theoretically confirmed by some other papers (Creath, K., & Schwartz, G. E. (2004). Measuring effects of music, noise, and healing energy using a seed germination bioassay. The Journal of Alternative & Complementary Medicine, 10(1), 113-122; Hassanien, R. H., Hou, T. Z., Li, Y. F., & Li, B. M. (2014). Advances in effects of sound waves on plants. Journal of Integrative Agriculture, 13(2), 335-348.)
Reviewer 2 Report
This manuscript introduces an interesting method of acoustic wave triggering for the interaction with the growth of plant biological systems. A statistically significant difference only for frequencies of 3–5 kHz is reported, compared to the treated with 300–500 Hz and the control. Other than acoustic frequency, acoustic pressure may need to be concerned and discussed. What is acoustic pressure acted on the seeds? Would acoustic pressure have influence on the germination?
I think the manuscript needs major revision before publication.
Author Response
This manuscript introduces an interesting method of acoustic wave triggering for the interaction with the growth of plant biological systems. A statistically significant difference only for frequencies of 3–5 kHz is reported, compared to the treated with 300–500 Hz and the control. Other than acoustic frequency, acoustic pressure may need to be concerned and discussed. What is acoustic pressure acted on the seeds? Would acoustic pressure have influence on the germination?
I think the manuscript needs major revision before publication.
Thanks to the reviewer for the suggestions. The manuscript has been enriched by adding some theoretical consideration on sound pressure levels and influence on the seeds in the designed experimental arrangement.
According to theoretical and experimental studies in the field (Hassanien, R. H., Hou, T. Z., Li, Y. F., & Li, B. M. (2014). Advances in effects of sound waves on plants. Journal of Integrative Agriculture, 13(2), 335-348), ‘Loudness’ is described as Sound Pressure Level (SPL) and it is measured in dB (decibels).
In order to determine an estimate of SPL at a distance, the Inverse Square Law can be used. It says that SPL decreases inversely with the square of the distance from the source at a rate of approximately 6 dB for each doubling of the distance: ???(?) = ???0 − 20???10(?)
where ? is the distance in meters, and ???0 is a sound pressure level at one meter from the source
Taking ???0 for zero, we can get the expression of change ??? for distance ? as ∆???(?) = −20???10(?).
Using this formula for d=1cm (0.01 meters):
SPL(d)=-20 Log10(0.01)=40 dB
According to this the influence of SPL on seeds in the experimental arrangement of this work is negligible. This data and the consequent considerations have been added in the revised text.
Round 2
Reviewer 1 Report
In the manuscript, “Micromachined Tools using Acoustic Wave Triggering for the Interaction with the Growth of Plant Biological Systems” was submitted for publication in the journal Micromachines. The manuscript is well revised according to the comments. I agreed that this manuscript can be accepted to the MEMS in Italy under the scope of Innovative applications of MEMS. I have only one comment after revision.
1. The results and data should be consistently presented in the manuscript. In Fig 15, there are error bars shown in the subfigure however there is no information that how many replicate data were obtained from the experiment. In other figures, there is no error bar at all. I suggest the author put all the error bars back to all the figures and make clear descriptions of how many replicate experiments are in each data.
Reviewer 2 Report
Well revised.